# Heat Transfer Enhancement inside Rectangular Channel by Means of Vortex Generated by Perforated Concave Rectangular Winglets



**Syaiful [1,\*], Monica Pranita Hendraswari [1], M.S.K. Tony S.U. [1] and Maria F. Soetanto [2]**

[1] Mechanical Engineering Department, Diponegoro University, Semarang 50275, Indonesia; monicapranitah@gmail.com (M.P.H.); msktonysu@yahoo.co.id (M.S.K.T.S.U.)

[2] Mechanical Engineering Department, Bandung State Polytechnic, Bandung 40012, Indonesia; mariasoetanto@polban.ac.id

\* Correspondence: syaiful@lecturer.undip.ac.id; Tel.: +62-812-2850-1462

**Abstract:** Passive methods using vortex generators (VGs) to enhance heat transfer have been a concern of researchers in recent decades. This study is intended to investigate the strength of the vortex generated by VGs by trying to reduce the pressure drop in the flow. The present work also takes into account the influence of the vortex intensity on the improvement of heat transfer, which can be indicated by the low value of the synergy angle. Experiments were carried out in the current investigation to validate the results of the numerical simulations in the Reynolds number range of 3102 to 16,132. The study results indicate that the observed heat transfer coefficients from the experimental and simulation results have a similar tendency with relatively small errors. A reduction in pressure drop is observed with the use of perforated concave rectangular winglets (PCRWs) against the nonperforated ones although there was a slight decrease in heat transfer improvements.

**Keywords:** vortex intensity; synergy angle; heat transfer; pressure drop





## 1. Introduction

The improvement of heat transfer in heat exchangers is significant for energy efficiency these days. Increased heat transfer in heat exchangers is more effective through a passive method using vortex generators (VGs). VGs generate longitudinal vortices, which enhance fluid mixing so that the heat transfer rate increases [1]. The longitudinal strength of the vortex (LV) is influenced by the aspect ratio of VG [2]. The increase in the aspect ratio reduced the strength of the vortex, according to their studies. Vortex is also able to overcome the weaknesses heat transfer rate in the behind of tube region by placing VGs in that area [3]. The LV produced by VG interacts with the boundary layer, enhancing the rate of heat transfer from the surface to the fluid [4]. Awais and Arafat studied numerically and experimentally the effects of the arrangement, location, and angle of attack of VG on the heat transfer characteristics and pressure drop of the flow [5]. Their work found that the delta winglet (DW) VG indicates better performance than the rectangular winglet (RW) VG.

The use of delta and rectangular winglets VGs to increase heat transfer rate in solar energy storage was carried out by Felipe et al. [6]. They observed that better thermal-hydraulic performance was found in the use of DW VGs with an angle of attack of 30°. From their results, however, the corner vortex was only generated from RW VGs. The vortex generated by the curved rectangular winglet vortex generator (RWVG) was capable of providing a suitable thermal-hydraulic performance than that of the RWVG wavy-up [7]. Their work indicated that the RWVG wavy-up yields the best heat transfer improvements. However, the j/f ratio shows a lower value than that of the curved RWVG. The best thermal-hydraulic performance was demonstrated by the curved arc winglet VG, as was found from the investigation results by Hui et al. [8]. They also observed that the heat

transfer improvement was better with curved arc VGs than RWVG. Improved heat transfer can be accomplished by combining vortex generators and dimple surfaces evaluated by Gaofeng and Xiaoqiang [9]. They also found that the use of VGs indicated the highest vortex intensity. The vortex generated behind the tube destroys the wake region, which results in better fluid mixing so that the heat transfer in the wake region becomes better [10].

Improved fluid mixing can be performed by evaluating the geometry and location of the VGs, including the number of VG mounted in the channel [11]. Their work showed a thermal improvement (TEF) with the addition of the RWVG. In terms of thermal enhancement, Li et al. proposed a novel VG mounted inside a helical channel [12]. They found that the VG streamlined winglet pair (SWP) improved heat transfer with the lowest pressure drop. The installation of VG is also a concern in increasing heat transfer. Arvind et al. evaluated the effect of the punched RWVG on the thermo-hydraulic performance of the fin and tube heat exchanger [13]. They informed that heat transfer was increased more by installing punched than non-punched RWVG. Heat transfer augmentation is always accompanied by an increase in pressure drop. Mingjie et al. considered the impact of increased pressure drop on pumping power to improve heat transfer by mounting a DWVG around the tube [14]. The thermal-hydraulic performance found by them was improved by installing VG around the tubes of the fin and tube heat exchanger. Pongjet and Sompol tried to compare the V shape arrangement of the rectangular and delta winglet VGs to the thermo-hydraulic performance [15]. They found that the V-shaped configuration for the delta winglet (DW) VG yielded a slightly higher TEF than that of the rectangular winglet (RW) VG.

Based on this literature study, VG can increase heat transfer by generating vortices that enhance fluid mixing. However, this improvement in heat transfer is accompanied by an increase in flow resistance, resulting in high pumping power. Therefore, the optimization of thermal-hydraulic performance has become a concern of many researchers recently. This optimization can be performed by reducing flow resistance while maintaining a high heat transfer rate [16,17]. Evaluating the use of a perforated concave rectangular VG to reduce flow resistance with increased heat transfer is extremely rare. Therefore, the present study focuses on a detailed analysis relating the flow pattern through VG to the increase in its heat transfer rate.

## 2. Materials and Methods

### 2.1. Experimental Set-Up

Detailed analysis of the effect of vortex dynamic on heat transfer improvement in this study was carried out by numerical simulation. In this study, experiments were also carried out to validate the numerical simulation results, as shown in Figure 1. This experiment was carried out in the mechanical engineering thermofluid laboratory of Diponegoro University. The channel for testing was made of glass with a thickness of 1 cm and a length, width, and height of 370 cm, 8 cm, and 18 cm, respectively. Air was sucked into the channel by a blower located at the end of the channel through a straightener with wire mesh behind, which functions to uniform the flow. Air velocity in the channel was varied in the range 0.4 m/s to 2.0 m/s with 0.2 m/s intervals. This air velocity was measured by a hot wire anemometer (Lutron type AM-4204 with an accuracy of ±0.1 m/s), which is located 27 cm behind the wire mesh. An inverter (Mitsubishi Electric-type FR-D700 with an accuracy of ±0.01) was used to regulate the flow velocity by controlling the blower's motor to obtain the desired flow velocity. Then, the flow passed through a plate (1 mm thick with length and width of 500 mm and 155 mm, respectively), which was heated constantly by a heater with/without VG. A test plate without VG attachment was termed the baseline. The constant heat induced into the plate was monitored by a wattmeter (Lutron DW-6060 with an accuracy of ±1.0) with a heating regulator setting. Several type K thermocouples were used to measure the inlet, outlet, and surface temperatures of the test plate. These thermocouples were connected to data acquisition (Advantech type USB-4718 with an accuracy of ±0.001) and linked to the CPU for monitoring and storage. The pressure drop

between the inlet and outlet points of the test plate was measured by two pitot tubes connected to a micromanometer (Fluke type 922 with an accuracy ±0.05).

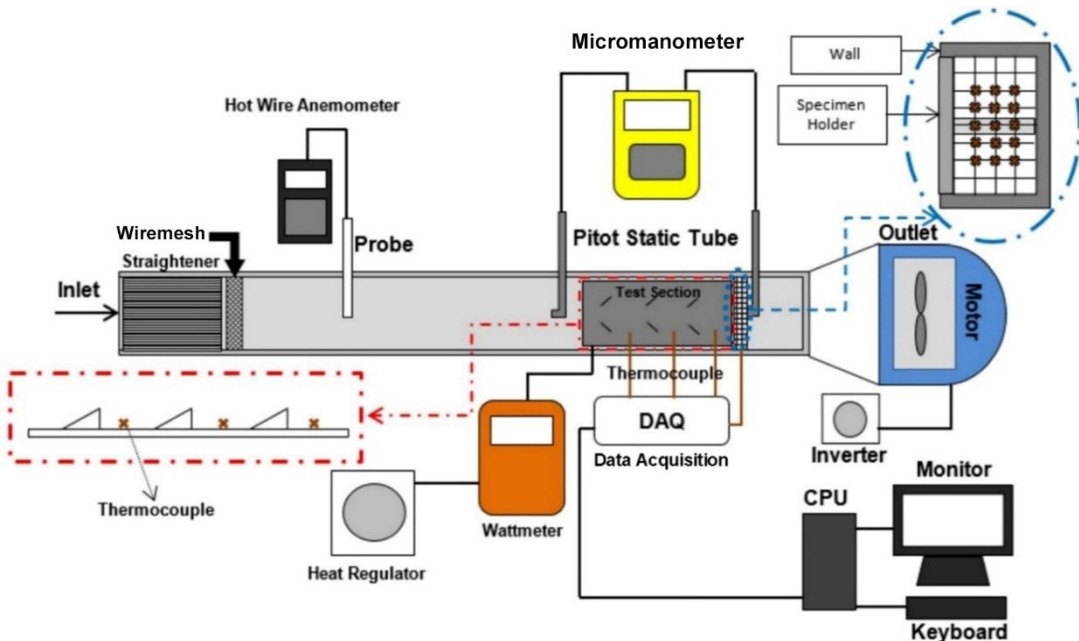

**Figure 1.** Schematic of experimental test equipment.

### 2.2. Physical Model

In this study, the VG used were concave rectangular winglet (CRW) and rectangular winglet (RW) arranged in common flow-down with an angle of attack of 15°. VG was made of the aluminum plate at a thickness of 1 mm with/without holes having a diameter of 5 mm. The dimensions and geometry of the CRW and RW VGs are shown in Figure 2. The longitudinal pitch distance between VGs is 125 mm, and the transverse pitch distance between a pair of VGs is 20 mm. Figure 3 illustrates a top view of the test plate with VGs. The detailed geometry of VG is presented in Table 1.

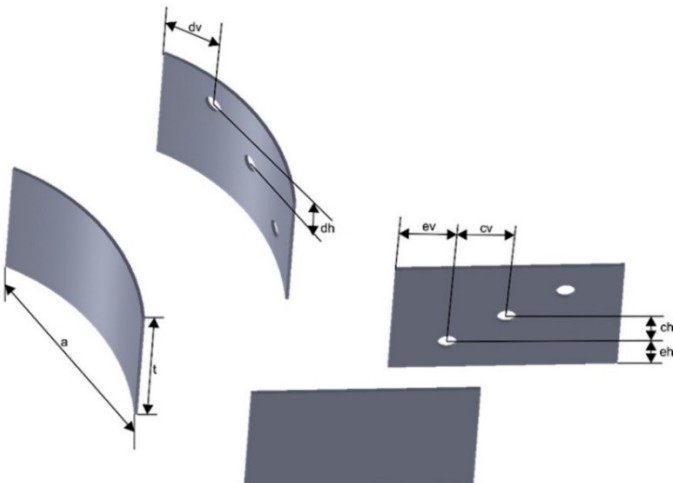

**Figure 2.** Detailed geometry of the concave rectangular winglet (CRW) and rectangular winglet (RW) vortex generators (VGs).

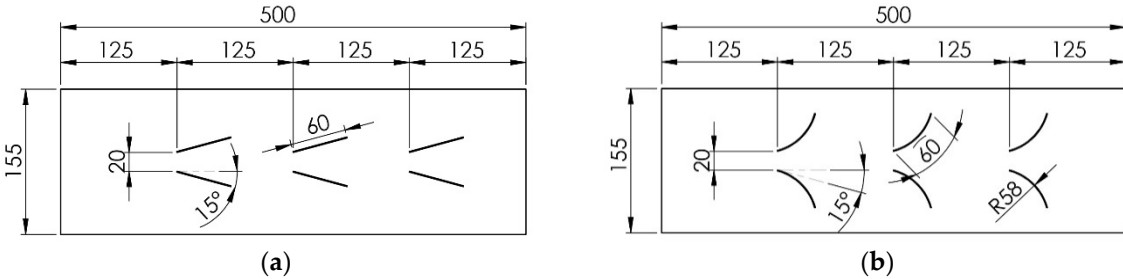

**Figure 3.** Top view of test plate with (**a**) RWVGs and (**b**) CRWVGs.

**Table 1.** Detail geometry of VG.

| VGs | α (º) | a (mm) | cv (mm) | dv (mm) | ev (mm) | ch (mm) | dh (mm) | eh (mm) | t (mm) | R (mm) |
|---|---|---|---|---|---|---|---|---|---|---|
| CRW without holes | 15 | 59 | - | - | - | - | - | - | 40 | 58 |
| CRW with holes | 15 | 59 | 29.56 | 44.56 | 14.56 | 20 | 30.15 | 9.85 | 40 | 58 |
| RW without holes | 15 | 60 | - | - | - | - | - | - | 40 | - |
| RW with holes | 15 | 60 | 30 | 45 | 15 | 20 | 30 | 10 | 40 | - |

The computational domain was determined from half the symmetrical volume of the channel volume where the test plate is located with the extension of the inlet and outlet sections, as shown in Figure 4. The x, y, and z axes show the direction of streamwise, spanwise, and normal wall, respectively. The extended inlet region (upstream extended region) is intended to obtain a fully developed flow on the inlet side of the test plate. The fluid region is the volume region of fluid with/without VG. The hot wall region is the region defined for conjugate heat transfer calculations. The extended outlet region (downstream extended region) is a region of extension on the outlet side which is intended to overcome reverse flow.

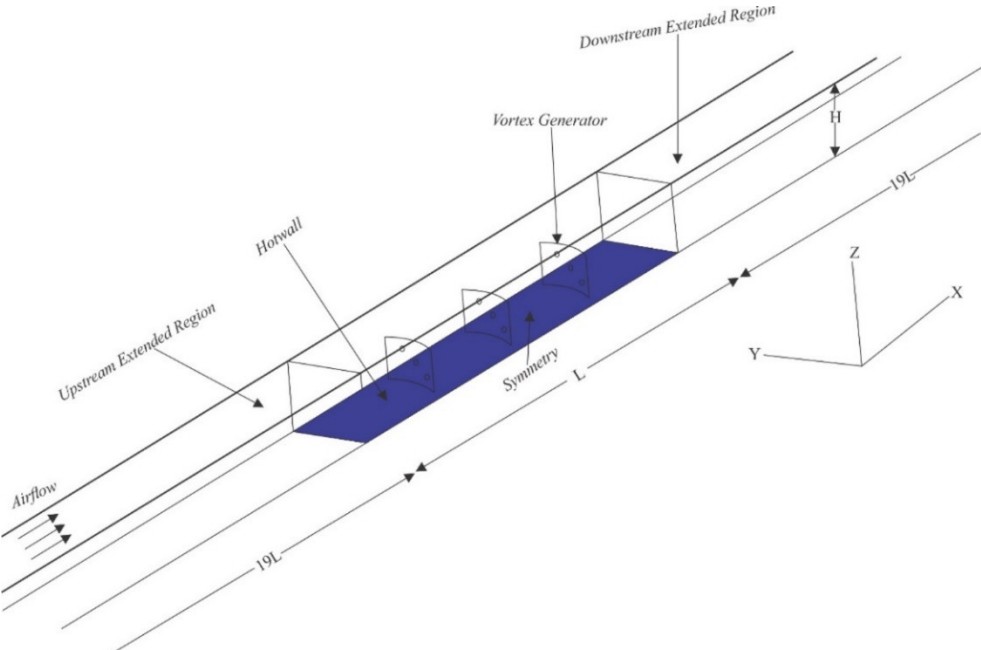

**Figure 4.** Computation domain.

### 2.3. Governing Equation

In the current study, the fluid flow was assumed to be steady-state, incompressible, and the physical properties of air were constant. Viscous dissipation and body force were

negligible. At air velocities of 0.4–2.0 m/s with 0.2 m/s intervals, the Reynolds number was in the range of 1600 to 9000. Therefore, fluid flow acts in a laminar and turbulent form. Based on the above assumptions, the governing equations applied in this modeling are as follows, [18]

Continuity equation,

$$\frac{\partial u_j}{\partial x_j} = 0 \tag{1}$$

Momentum equation,

$$\frac{\partial}{\partial x_j}(\rho u_i u_j) = -\frac{\partial p}{\partial x_i} + \frac{\partial}{\partial x_j}\left(\mu \frac{\partial u_k}{\partial x_i}\right) \tag{2}$$

Energy equation,

$$\frac{\partial}{\partial x_i}(\rho u_i T) = \frac{\partial}{\partial x_i}\left(\Gamma \frac{\partial T}{\partial x_i}\right) \tag{3}$$

where $\rho$, $u$, $p$, $\mu$, $T$, and $\Gamma$ are density, mean velocity on the x-axis, pressure, dynamic viscosity of the air, temperature, and the diffusion coefficient, respectively. The diffusion coefficient can be stated as $\Gamma = \lambda/c_p$ where $\lambda$ and $c_p$ are the thermal conductivity and specific heat of the fluid, respectively. In this modeling, the $k$-$\omega$ standard was used to simulate the turbulent model, which is stated as follows,

$$\frac{\partial}{\partial x_i}(\rho k u_i) = \frac{\partial}{\partial x_j}\left(\Gamma_k \frac{\partial k}{\partial x_j}\right) + G_k - Y_k + S_k \tag{4}$$

$$\frac{\partial}{\partial x_i}(\rho \omega u_i) = \frac{\partial}{\partial x_j}\left(\Gamma_\omega \frac{\partial \omega}{\partial x_j}\right) + G_\omega - Y_\omega + S_\omega \tag{5}$$

where $\Gamma_k = \mu + \mu_t/\sigma_k$ and $\Gamma_\omega = \mu + \mu_t/\sigma_\omega$. The turbulent intensity was determined using the following correlation:

$$I = 0.16 Re_{D_h}^{-1/8} \tag{6}$$

### 2.3.1. Boundary Conditions

Boundary conditions are required to solve governing equations. The boundary conditions defined in the computational domain are stated as follows [16];

Inlet upstream extended region:

$$u = u_{in}, \ v = w = 0, \ and \ T = T_{in} = const. \tag{7}$$

Outlet downstream extended region:

$$\frac{\partial u}{\partial x} = \frac{\partial v}{\partial x} = \frac{\partial w}{\partial x} = 0, \ \frac{\partial T}{\partial x} = 0 \tag{8}$$

Wall:

$$u = v = w = 0, \ and \ q = 0 \tag{9}$$

Hot wall:

$$u = v = w = 0, \ and \ T = T_w \tag{10}$$

Symmetry:

$$v = 0, \ \frac{\partial u}{\partial y} = \frac{\partial w}{\partial y} = \frac{\partial T}{\partial y} = 0 \tag{11}$$

### 2.3.2. Numerical Method

Computational fluid dynamic was determined by dividing the control volume from the computational domain into small control volumes which are termed mesh generation.

In order to obtain accuracy in modeling, two mesh types were proposed, namely, the tetrahedral and the hexahedral mesh. Hexahedral mesh was implemented for upstream and downstream extended regions because of their simpler geometry. Tetrahedral mesh was used in region where VGs are mounted to the hot plate. This mesh generation can be observed in Figure 5.

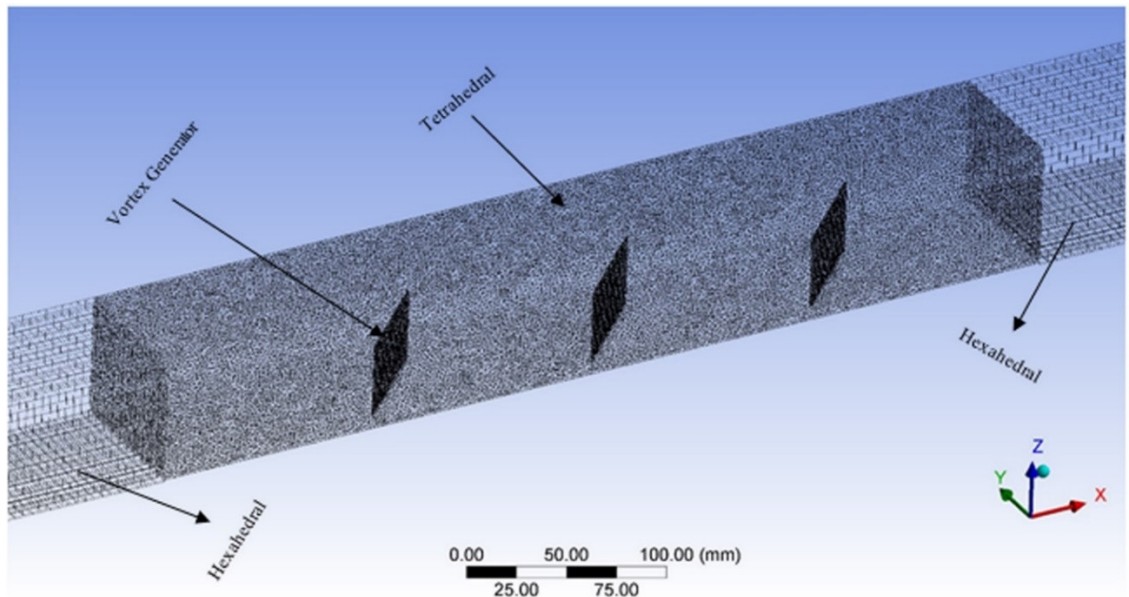

**Figure 5.** Mesh Generation.

An independent grid test was performed to ensure the calculation results are not affected by the grid number. Since the small difference in the convection coefficient (h) value, the independent grid was determined based on the small error between h simulation and experimental results, as indicated in Table 2. Based on the results of the independent grid test, it was found that the number of grids ranging from 1,600,000 was selected in this numerical simulation because of the low error.

**Table 2.** Grid independent test.

| Element Number | $h_{\mathrm{simulation}}$ (W/m$^2$·K) | $h_{\mathrm{experiment}}$ (W/m$^2$·K) | Error (%) |
|---|---|---|---|
| 1,262,840 | 18.27726 | 18.18571 | 0.503 |
| 1,478,060 | 18.34781 | 18.18571 | 0.891 |
| 1,661,610 | 18.24699 | 18.18571 | 0.337 |
| 1,868,587 | 18.29429 | 18.18571 | 0.597 |

In this work, computational fluid dynamics (CFD) with the finite volume method was used to analyze the thermo-hydraulic characteristics in the channel. In this study, the flow was modeled as laminar flow at low velocity, and the rest was modeled as turbulent flow. The turbulent k-ω standard model was applied in this work because this model is suitable for modeling fluid flow in the viscous layer [19]. The numerical solution of the continuity and momentum equations was determined using the SIMPLE algorithm. The equations for momentum, turbulent kinetic energy, specific dissipation rate, and energy were discretized using a second-order upwind scheme. The convergence criteria for the continuity, momentum, and energy equations were determined $10^{-5}$, $10^{-5}$, and $10^{-8}$, respectively.

### 2.4. Model Validation

To obtain accurate simulation results, the modeling results were validated with the results of experiments conducted in the Thermofluid Laboratory of Diponegoro University. The results of the experiments conducted were also compared with the work of Wu and Tao (2008), and this validation has been published previously (see Ref. [16]). From the experimental validation results, Syaiful et al. (2019) and Wu and Tao (2008) found a deviation. This deviation is due to differences in flow velocity and heat rate direction, which is the heat rate in the experiment of Syaiful et al. (2019) only goes to one direction because the other direction is isolated, while Wu and Tao (2008) go two ways.

## 3. Results and Discussion

### 3.1. Velocity Vector and Streamline

Figure 6 shows the flow pattern in a channel with/without VG. From Figure 6, it is found that swirl flow is formed behind VG. Swirl flow, or, in other words, longitudinal vortex (LV), is not observed in the baseline case. LV is produced by VG due to flow separation along VG caused by the pressure difference between the upstream and downstream regions of VG [20]. The strongest swirl flow is found when the fluid passes through CRWP VG, as shown in Figure 6. Counter-rotating LV produced by RWP (rectangular winglet pair) and CRWP (concave rectangular winglet pair) VG causes fluid to move into the hot wall, as can be seen in Figure 7. Strong counter-rotating pairs of LV form behind the VG with the left rotating clockwise and the right turning counterclockwise. LV counter-rotating pairs in the common-flow down configuration cause fluid to move away from the bottom wall (hot wall) and towards the upper wall, which is termed the up-wash region. Meanwhile, the fluid moving towards the bottom wall (hot wall), which is observed between the two vortex cores, is referred to as the down-wash region (see Ref [21]).

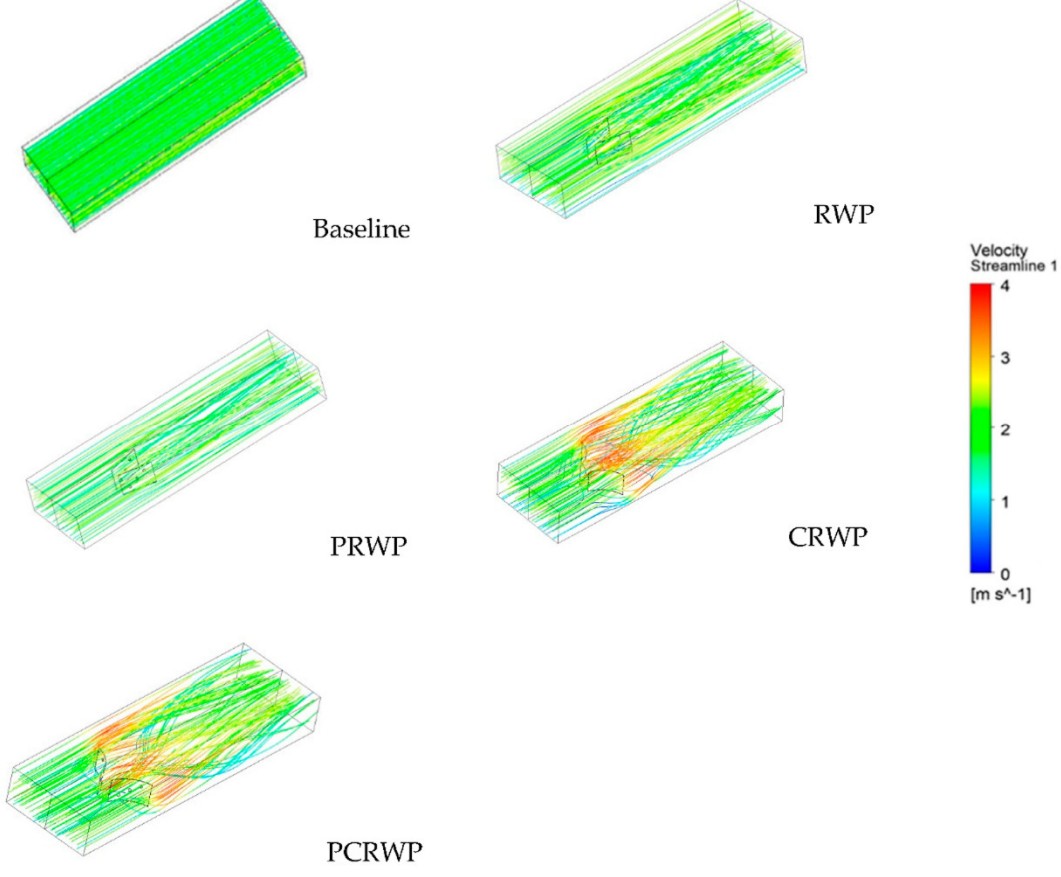

**Figure 6.** Streamline velocity of flow with/without VG.

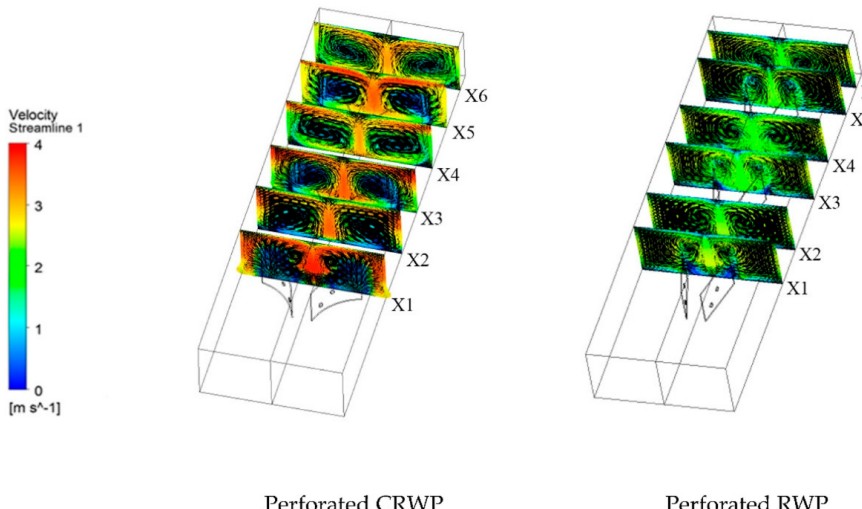

Perforated CRWP                    Perforated RWP

**Figure 7.** Tangential velocity vector on the use of perforated RWP and CRWP VGs.

In order to observe in detail, the tangential velocity vectors in RWP and CRWP installations with/without holes, the cross-sectional area of the flow at location X1 is selected, as shown in Figure 8. The fluid velocity, in this case, is 2.0 m/s. Figure 8 shows that the tangential velocity vector is higher in the downwash area resulting in increased local heat transfer (see Ref. [5]). The strength of the LV generated by the CRW shows that the CRW has a larger LV than the RWP because of the instability of the flow due to centrifugal force when the fluid passes through CRW VG [22]. The holes in VG result in a jet flow formation which can remove stagnant fluid and reduce the pressure difference before and after VG [23]. This decrease in pressure difference causes a reduction in LV strength. However, jet flow formation can reduce the wake region, which positively impacts in increasing heat transfer.

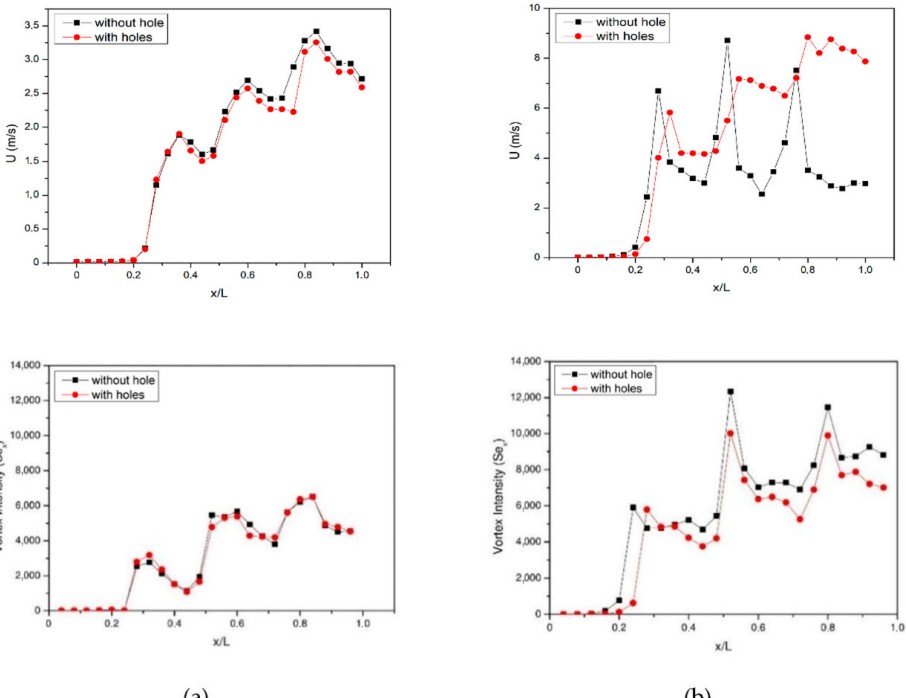

(a)                                    (b)

**Figure 8.** Tangential velocity and local vortex intensity for (**a**) RW VG and (**b**) CRW VG at the flow velocity of 2 m/s.

### 3.2. Longitudinal Vortex Intensity

In this study, a dimensionless number is used as a representation of the strength of the LV to determine the quantitative comparison of the LV intensity defined in Equation (12) [24],

$$Se = \frac{\rho D_h U}{\mu} \tag{12}$$

*Se* is a dimensionless number that represents the inertia force induced by secondary flow to the viscous force. *U* is the secondary flow velocity characteristic which can be formulated by Equation (13), [24]

$$U = D_h|\omega^n| = D_h\left|\frac{\partial w}{\partial y} - \frac{\partial v}{\partial z}\right| \tag{13}$$

where $\omega^n$ is the vorticity about the normal axis of the cross-section area. LV intensity is defined as follows, [24]

$$Se_x = \frac{\rho D_h^2}{A(x)\mu} \iint\limits_{A(x)} |\omega^n| dA \tag{14}$$

In the case of CRW and RW VG, the LV intensity tends to be dissipated after passing VG at the position x/L = 0.4 due to the viscous effect [25], which causes a decrease in LV intensity, as can be seen from Figure 8 at a flow rate of 2 m/s. Lis the length of the plate to which VG is mounted, while x is the position of the cross-section of the inlet section. The LV intensity in the CRW VG case is observed to be higher than that of the RWP due to flow instability caused by centrifugal force when the flow passes through CRW VG [22]. In the use of RW and CRW VGs, the holes on the VG result in a slightly lower LV intensity than those without holes. The main reason for this is that the hole in VG causes a weakening of the strength of the LV so that the intensity of mixing between cold fluid and hot fluid decreases [16]. In the perforated RW VG with a flow velocity of 2.0 m/s, the LV intensity increased by 3.92% compared to RW VG without holes at x/L = 0.8. Whereas in the case of perforated CRW VG at the same flow velocity, the LV intensity also increases by 2.88% than that of CRW VG without holes at x/L = 0.8.

### 3.3. Temperature Distribution

The flow pattern due to VG installation impacts the temperature distribution, as can be seen in Figure 9. Figure 9 shows the temperature distribution in the cross-section area at several locations for all cases. Overall, the temperature distribution in the cross-section area at several locations with VG mounted is better distributed than without VG (baseline). CRW VG is distributed better than RW VG because of the distortion in the thermal boundary layer due to a strong swirling flow, which causes damage to the thermal resistance between the walls and the flow so that heat transfer increases [25]. Figure 9 shows that the hole in VG causes the LV intensity to decrease due to the formation of jet flow which results in a slight reduction in the temperature gradient [23].

### 3.4. Pressure Distribution

Figure 10 shows the pressure distribution in the area streamwise in a channel mounted with VG for the flow velocity of 2.0 m/s. The installation of VG in the channel obstructs airflow, which impacts on increasing pressure drop due to the formed drag in the flow [26]. Figure 10 shows that CRWP VG results in a greater pressure drop than that of RWP VG because the frontal area of CRWP VG is larger than that of RWP VG. In addition, the holes in VG reduce pressure drop because jet flow removes stagnant fluid in the downstream region of VG, reducing the pressure difference before and after VG [23].

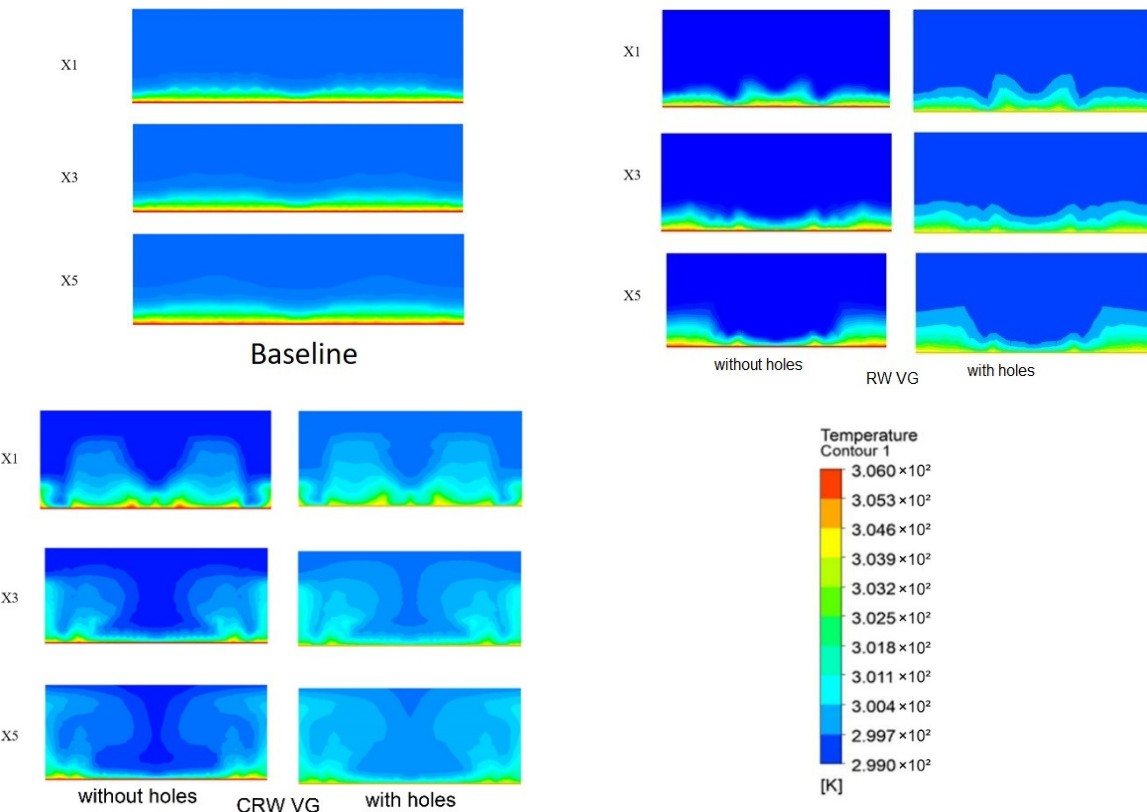

**Figure 9.** Temperature distribution in the cross-section area at several locations for all cases.

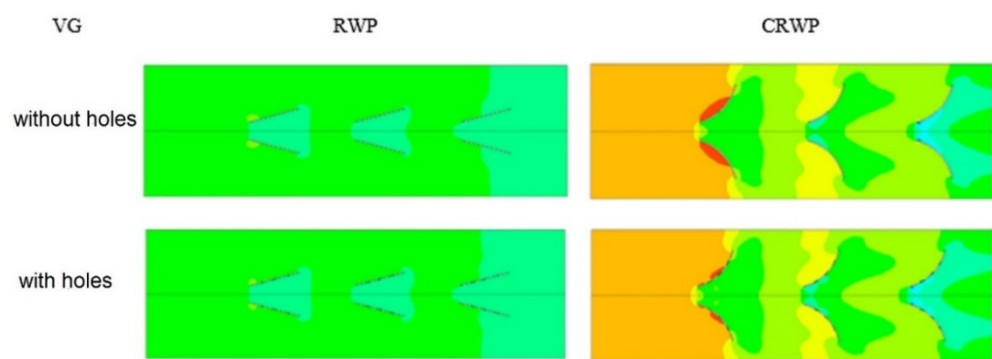

**Figure 10.** Pressure distribution in streamflow direction with mounting VG.

### 3.5. VG Impacts on Average Local Nusselt Number

The improvement of local heat transfer to the flow in the channel can be observed through the mean spanwise Nusselt number as promoted by Hiravennavar et al. [27].

$$\overline{Nu_s} = \frac{Bq(H/k)}{\int_0^B (T_w - T_b)dz} \tag{15}$$

where $B$ is the channel width, $q$ is the heat flux, $H$ is the channel height, and $k$ is the thermal conductivity of the fluid. At the same time, $T_w$ and $T_b$ are the wall temperature and bulk fluid temperature, respectively.

Figure 11 shows the comparison of the mean spanwise Nusselt numbers and energy dissipation rate in the RW and CRW VG cases with or without holes at a flow velocity of 2.0 m/s. From Figure 11, it can be observed that the mean spanwise Nusselt and energy

dissipation rate in the use of CRW VG is higher than that of RW VG due to the higher intensity of LV in the case of CRW VG, as illustrated in Figure 8. The holes in VG slightly reduce the strength of the LV, resulting in a decrease in the mean spanwise Nusselt number and energy dissipation rate, as observed in Figure 11. The highest average spanwise Nusselt numbers in the case of RW and CRW VG without holes at flow velocity are 5.58 at x/L = 0.32 and 0.61 at x/L = 0.38, respectively.

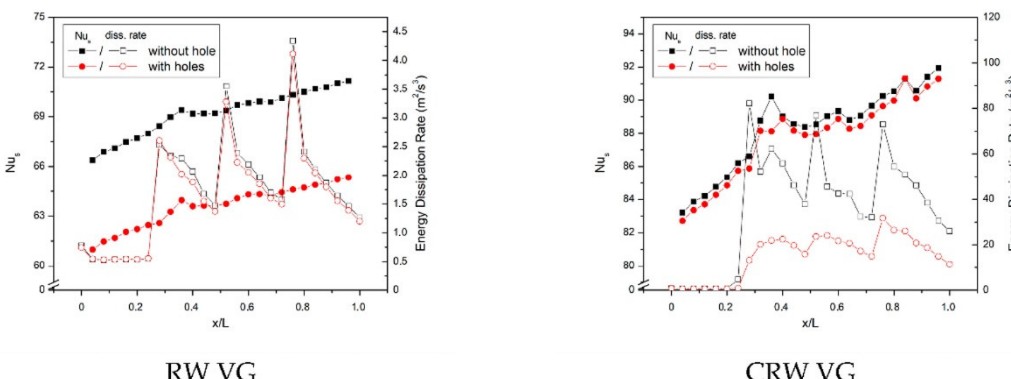

**Figure 11.** Local average spanwise Nusselt number and energy dissipation rate at the flow velocity of 2.0 m/s.

### 3.6. VG Effects on Heat Transfer Rate

Figure 12 describes the comparison of the convection heat transfer coefficient values from the experimental and simulated results using the RW and CRW VGs with/without holes and the baseline. Figure 12 indicates that the CRW VG installation results in a higher convection heat transfer coefficient than RW VG at the same flow velocity. The main reason for this is that the fluid experiences centrifugal force instability when the fluid passes through the concave wall of the CRW VG which results in a stronger LV intensity than RW VG [16]. The convection heat transfer coefficient slightly decreases with the holes in VG caused by the weakening of the LV due to jet flow so that the mixing intensity of hot and cold fluids reduces. At the highest flow velocity, the convection heat transfer coefficient for perforated VG decreases by 1.02% and 4.06% from that for without holes in CRW and RW VGs, respectively.

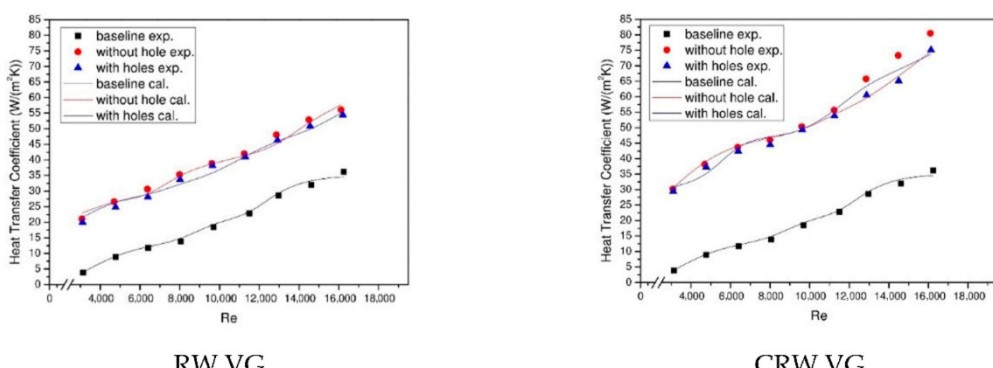

**Figure 12.** Comparison of convection heat transfer coefficient between with and without holes for installation of VG as well as baseline.

### 3.7. VG Effects on Flow Pressure Drop

The holes in VG impact the convection heat transfer coefficient and also on the pressure drop of the flow. Comparison of pressure drop for the use of RW and CRW VGs with holes and without holes from the experimental and simulation results is shown in Figure 13.

Figure 13 shows that the use of CRW VG results in a higher pressure drop than RW VG because of the larger frontal area of CRW VG resulting in high flow resistance [16]. The use of perforated RW and CRW VGs reduces pressure drop because the LV produced by VG is slightly weakened resulting in flow leakage. The holes in the VG reduced the pressure drop by about 15.38% and 7.69% for the installation of CRW and RW VGs, respectively.

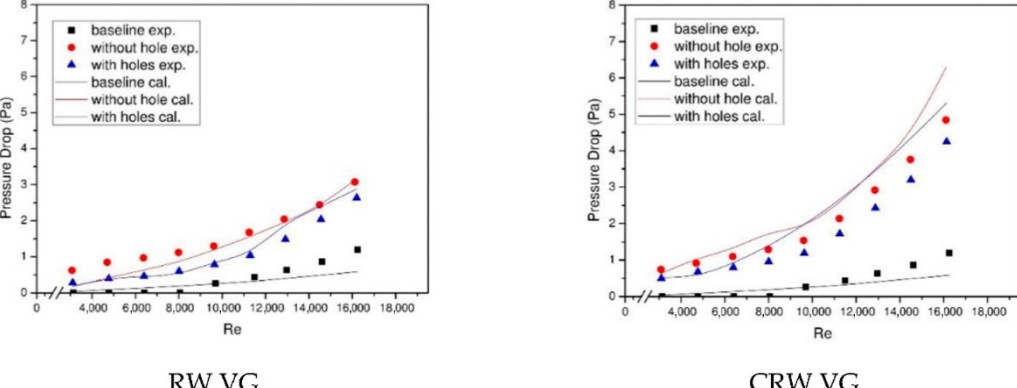

RW VG                                           CRW VG

**Figure 13.** Comparison of flow pressure drop between RW and CRW VGs use with/without holes as well as baseline.

### 3.8. Field Synergy Principe Analysis

FSP is a method for analyzing the improvement in heat transfer rates, as promoted by Guo et al. [28]. This method is based on the fluid energy balance near the surface, which is determined as follows,

$$\rho c_p \int_0^{\delta_t} (U \cdot \nabla T) dy = -\lambda \frac{\partial T}{\partial y} \tag{16}$$

where $\rho$, $c_p$, and $\lambda$ are assumed to be constant and in dimensionless form, Equation (16) is of the form:

$$Re_x Pr \int_0^1 \left( \vec{U} \cdot \vec{\nabla T} \right) d\vec{y} = Nu_x \tag{17}$$

where $\vec{U} = \frac{U}{U_\infty}$, $\vec{\nabla T} = \frac{\nabla T}{(T_\infty - T_w)/\delta_t}$, $\vec{y} = \frac{y}{\delta_t}$. $U_\infty$ and $T_\infty$ are the velocity and temperature of the fluid in the freestream, respectively. $\delta_t$ is the thickness of the thermal boundary layer. Synergy angle is the angle between velocity vector and temperature gradient of the flow $(\vec{U} \cdot \vec{\nabla T})$. Equation (17) revealed that $Nu_x$ can be increased by increasing $\vec{U} \cdot \vec{\nabla T}$ which can be expressed as $\vec{U} \cdot \vec{\nabla T} = \left| \vec{U} \right| \left| \vec{\nabla T} \right| cos\theta$. The increase of $\vec{U} \cdot \vec{\nabla T}$ can be obtained by reducing synergy angle $(\theta)$.

Figure 14 illustrates the synergy angle comparison between the RW and CRW VGs cases with and without holes at a flow velocity of 2 m/s. Figure 14 shows that the use of CRW VG reduces the synergy angle greater than RW VG because CRW VG has a stronger LV which can reduce the synergy angle [23]. The mean synergy angles found for the RW and CRW VG in the presence of holes are 0.41° and 0.25°, respectively, at a flow velocity of 2.0 m/s. From Figure 14 it can be clearly observed that the synergy angle of perforated VG is slightly higher than that of VG without holes.

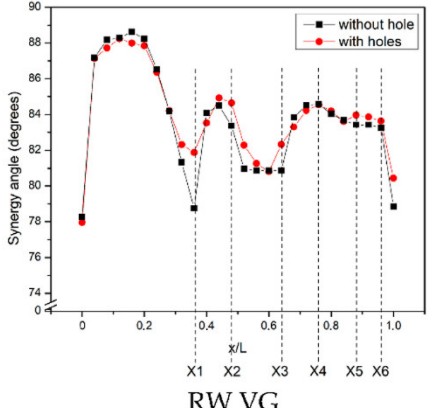
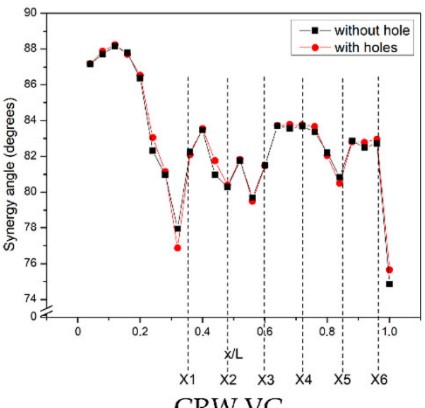

**Figure 14.** Synergy angle for RW and CRW VG use with and without holes.

## 4. Conclusions

Three-dimensional numerical simulation of airflow through rectangular ducts fitted with RW and CRW VGs with and without holes at flow velocity with angle of attack 15 was performed. From the simulation results, it can be concluded from this work as follows:

1. The decrease in the convection heat transfer coefficient in the case of perforated CRW VG was 1.02% of the CRW VG without holes at a flow velocity of 2.0 m/s. Whereas in a similar case for RW VG, the convection heat transfer coefficient decreased by 4.06% from that of RW VG without holes at the same flow velocity.
2. Pressure drop on perforated VG decreased by 15.38% in cases of CRW VG with holes and 7.69% in cases of RW VG with holes at the highest flow velocity.
3. The average synergy angles in the RW VG case are 0.41° and 0.25° in the CRW VG for the highest flow velocity.

**Author Contributions:** Create concepts and methodologies from studies, S.; Simulating the problem, M.P.H.; Check simulation results and validation, M.S.K.T.S.U.; Perform experiments to validate simulation results, M.F.S. All authors have read and agreed to the published version of the manuscript.

**Funding:** This work is funded by the tertiary superiority basic research program (PDUPT) of RISTEK DIKTI (Minister of Education) Indonesia with the grant number: 225-110/UN7.6.1/PP/2020.

**Data Availability Statement:** The data that I provide in this paper is original.

**Acknowledgments:** This work was supported by the Ministry of Education and Culture, Indonesia, with contract number: 225-110/UN7.6.1/PP/2020. The authors are grateful to all research members, especially Lab. of Thermofluid of Mechanical Engineering of Diponegoro University Indonesia, Lab. of Aerospace Engineering Department of Bandung State Polytechnic, Indonesia.

**Conflicts of Interest:** The authors declare no conflict of interest.

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
