# Peer review of "Heat Transfer Enhancement inside Rectangular Channel by Means of Vortex Generated by Perforated Concave Rectangular Winglets"

_fluids, doi:10.3390/fluids6010043_

Round 1
Reviewer 1 Report
Review
of the Manuscript
"Effect of Vortex Dynamics Generated by Perforated Concave Rectangular Winglet Vortex Generators On Heat Transfer Enhancement Inside Rectangular Channel"
of the authors:
Syaifu, Monica Pranita Hendraswari, MSK Tony SU and Maria F. Soetanto
The paper is quite fairly written and will be interesting for the readers.
- Originality
The article is novel and interesting and adheres to the journal's standards.
- Structure
The article is laid out quite clearly and fairly written.
All key elements are presented.
The figures and tables do inform the reader, and they are an important part of the story. However, some issues should be fixed (see comments below).
- Previous Research
The article builds upon previous research and makes references to those works appropriately.
Some remarks and comments:
Minor:
1) Fig. 3. It would be better if you show the size 20 mm between two adjacent plates of VG.
2) What means the name Syaiful (is it a first name or surname)? What is MSK? Maybe it means M.S.K.?
Major:
1) The title of the paper is extremely long. I believe one can drop off the word “Effect” and to avoid double use of “Generated” and “Generators”, “Vortex” in the title.
E.g. “Heat Transfer Enhancement Inside Rectangular Channel by means of Vortex Generated by Perforated Concave Rectangular Winglets”
2) Abstract. What means “low value of the synergy angle”? Is this an angle between what?
3) Abstract. “A reduction in pressure drop is observed with the use of perforated concave rectangular winglets (PCRWs) against the nonperforated ones”.
This result should be checked in the light of the relation between obtained effect (e.g. heat transfer coefficient) against speeded energy dissipation rate (W/kg or W/m3). see also comment 9.
4) Fig. 1. What means “Pressure micromanometer”? Use either “Differential pressure gauge” or “micromanometer”.
5) Fig. 1. What is DAQ? It would be nice to add a list of abbreviations.
6) Section 2.3. Please add the references for all the equations presented here.
7) Lines 131-132. “The diffusion coefficient can be stated as Gamma= Lambda/Cp”. Is it thermal diffusivity? It looks like thermal diffusivity multiplied by the density.
8) Eq. 8. dv/dx and dT/dx do not have the same units.
9) The comparison between RW VG and CRW VG, both with and without holes (figs. 10, 11 and 13) should be supplemented by the plots Nu = f(energy dissipation rate).
10) Fig. 12: it is not clear what for belong left part and right part. Please assign them as (a) and (b) and provide information in the figure caption.
11) The eq. (12) contains the secondary flow velocity characteristic U, calculated as given by eq. (13). Please add the plot showing the rate of U for all four cases (RW VG and CRW VG, both with and without holes).
12) Fig. 10 “with mounting VG” should be replaced by “with mounted VG”.
13) Some extended comment concerning the Field synergy principle (not a “principe”!) should be given, especially to the units (degrees) presented in Fig. 14.
Besides, the Fig. 14 should be well commented. The positions X1- X6 should be shown there.
14) Conclusion: “angle of attack 15”. Please add the angular units (degrees).
The paper could be accepted but needs some corrections.
Author Response
I thank reviewer 1 for the review. Your review provides an increase in my knowledge of what I need to be analyzed and discussed.
I've already revised everything you asked for, as I attach it in the reviewer file.

Reviewer 2 Report
In the present study, vortex generators (VGs) were used to enhance heat transfer in a rectangular channel. Both perforated and non-perforated flat and concave VGs were used. The studies were mainly conducted through numerical simulations, with experimental studies as the verifications. The effects of VGs on velocity field, vortex intensity, temperature distribution, pressure distribution, Nusselt number, pressure drop and Synergy angle were examined.
It seemed that the studies were conducted with great care and the results are new and interesting. The discussion is quite extensive. I have only some minor suggestions and comments. After the paper is revised by considering the following points, I am happy to recommend publication in Fluids.
- The title of the paper seems not reflect the studies appropriately. I thought that the paper focused mainly on “Perforated Concave Rectangular Winglet Vortex Generators”. But actually this is not the case. I would suggest a more general title rather than focus only on “Perforated VGs”.
- Still in the title, “Vortex Dynamics” seems mainly to refer to longitudinal vortices in the present study. I suggest the author make the title shorter. Also, it should be “inside a rectangular channel.
- The Abstract and Conclusions. After going through the Abstract and the Conclusion, I could not figure out clearly what are the main findings of this study. I strongly suggest the authors rewrite them to emphasize their main results in terms of the effect of different VGs on heat transfer enhancement.
- The format of the cited references in the text. The authors need to refer to the format of reference citations in FLUIDS. It seems a bit awkward to me to cite both the first name and the surname of a reference in the text.
- On page 2 about the accuracy of the hot wire anemometer. It is ±1 m/s. As the velocity range in the channel is from 0.4m/s to 2m/s, this accuracy seems a bit low. Can the author provide some comments on this?
- Page 3, what is the unit for Fluke type 922?
- Page 7, typing errors for 10-5.
- The definition for the synergy angle needs to be provided.
- In Conclusions, 1.02% was cited. This difference is quite small and could be smaller than the accuracy of the method used in this study. Do the author have any comments on this?
Author Response
I thank you for the review of my paper. I have revised the paper, as your suggestion. I attached the revised paper and my answer to your comment.

Round 2
Reviewer 1 Report
The authors have fixed all the issues raised by the reviewer. The paper could be accepted now.